# Fecal microbiota transplantation from protozoa-exposed donors downregulates immune response in a germ-free mouse model, its role in immune response and physiology of the intestine

Oswaldo Partida-Rodríguez[1,2], Eric M. Brown[2], Sarah E. Woodward[2], Mihai Cirstea[2], Lisa A. Reynolds[2,3], Charisse Petersen[2], Stefanie L. Vogt[2], Jorge Peña-Díaz[2], Lisa Thorson[2], Marie-Claire Arrieta[2,4], Eric G. Hernández[1], Liliana Rojas-Velázquez[1], Patricia Moran[1], Enrique González Rivas[1], Angélica Serrano-Vázquez[1], Horacio Pérez-Juárez[1], Javier Torres[5], Cecilia Ximénez[1]*, B. B. Finlay[2,6,7]

1 Unidad de Investigación en Medicina Experimental, Hospital General de Mexico, Universidad Nacional Autónoma de México, Mexico, Mexico, 2 Michael Smith Laboratories, Department of Microbiology & Immunology, University of British Columbia, Vancouver, Canada, 3 Department of Biochemistry and Microbiology, Faculty of Science, University of Victoria, Victoria, Canada, 4 Department of Physiology and Pharmacology, Cumming School of Medicine, University of Calgary, Calgary, Canada, 5 Unidad de Investigación en Enfermedades Infecciosas y Parasitarias, Instituto Mexicano del Seguro Social (IMSS), Mexico, Mexico, 6 Department of Microbiology and Immunology, Faculty of Science, University of British Columbia, Vancouver, Canada, 7 Department of Biochemistry and Molecular Biology, Faculty of Medicine, University of British Columbia, Vancouver, Canada

* cximenez@unam.mx

## Abstract

Intestinal parasites are part of the intestinal ecosystem and have been shown to establish close interactions with the intestinal microbiota. However, little is known about the influence of intestinal protozoa on the regulation of the immune response. In this study, we analyzed the regulation of the immune response of germ-free mice transplanted with fecal microbiota (FMT) from individuals with multiple parasitic protozoans (P) and non-parasitized individuals (NP). We determined the production of intestinal cytokines, the lymphocyte populations in both the colon and the spleen, and the genetic expression of markers of intestinal epithelial integrity. We observed a general downregulation of the intestinal immune response in mice receiving FMT-P. We found significantly lower intestinal production of the cytokines IL-6, TNF, IFN-γ, MCP-1, IL-10, and IL-12 in the FMT-P. Furthermore, a significant decrease in the proportion of CD3+, CD4+, and Foxp3+ T regulatory cells (Treg) was observed in both, the colon and spleen with FMT-P in contrast to FMT-NP. We also found that in FMT-P mice there was a significant decrease in tjp1 expression in all three regions of the small intestine; ocln in the ileum; reg3γ in the duodenum and relmβ in both the duodenum and ileum. We also found an increase in colonic mucus layer thickness in mice colonized with FMT-P in contrast with FMT-NP. Finally, our results suggest that gut protozoa, such as *Blastocystis hominis*, *Entamoeba coli*, *Endolimax nana*, *Entamoeba histolytica/E. dispar*, *Iodamoeba bütschlii*, and *Chilomastix mesnili* consortia affect the immunoinflammatory state and induce

**Data Availability Statement:** All relevant data are within the manuscript and its Supporting Information files.

**Funding:** This work was partially funded by the PAPIIT program at the Universidad Nacional Autónoma de México UNAM (grant numbers IN226511, IN218214, and IN217821) awarded to C.X., Instituto Mexicano del Seguro Social (IMSS) (grant FIS/IMSS/PROT/1368), awarded to J.T., the National Consejo Nacional de Ciencia y Tecnologia (CONACyT); grants numbers 140990, 272601, 283522, and 257091) awarded to C.X. and J.T., and the Canadian Institutes for Health Research awarded to B.B.F., "Estancias Posdoctorales en el Extranjero para la Consolidación de Grupos de Investigación" program of CONACyT (proposal number 208253) awarded to O.P.-R. The funders had no role in study design, data collection, and analysis, the decision to publish, or preparation of the manuscript.

**Competing interests:** The authors have declared that no competing interests exist.

functional changes in the intestine via the gut microbiota. Likewise, it allows us to establish an FMT model in germ-free mice as a viable alternative to explore the effects that exposure to intestinal parasites could have on the immune response in humans.

## Introduction

Intestinal infections due to parasitic protozoans represent a key component of the global burden of infectious diseases, constituting a major public health problem in most underdeveloped countries [1]. Both pathogenic and commensal intestinal parasites are part of the complex intestinal ecosystem [2, 3]. Parasites depend on their hosts to thrive, and some of them can produce metabolites that disrupt their hosts' microbiota and homeostasis and activate pathological aspects of the host immune response [4].

During intestinal parasite-associated diseases, such as amoebic colitis and amoebic liver abscess (both caused by *Entamoeba histolytica*) or intestinal giardiasis (caused by *Giardia duodenalis* infection), the direct associations of these parasites with intestinal bacteria can potentially influence parasitic or bacterial pathogenicity [5–8]. For instance, decreased abundances of *Bacteroides*, *Clostridium*, and *Lactobacillus* and the increased abundance of *Bifidobacterium* are associated with amoebic colitis [5], and the increased abundance of *Prevotella copri* is associated with amoebic diarrhea [6].

Although, pathogenic intestinal parasitic protozoans are the most studied, nonpathogenic protozoans have also been identified as having the capacity to alter intestinal microbiota diversity and stimulate host immune responses. However, relatively little research has examined gastrointestinal tract immunity generated by the presence of protozoa that exist as non-pathogenic in the gut [9]. Non-pathogenic parasitic protozoa have been shown to establish close interactions with the intestinal microbiota [10–12]. Beyond their role in digestion and nutrient acquisition, the presence of commensal parasitic protozoans is fundamental to maintaining intestinal homeostasis and modulating the development and maturation of the immune system, which is pivotal to effectively protect the host against other pathogenic microorganisms [13].

The intestinal parasitic protozoa *Blastocystis* is one of the non-pathogenic parasites more studied and is commonly found as a member of a healthy microbiota [3, 14, 15]. Additionally, there is evidence that the presence of this parasite can directly influence the composition of the gut bacteriome [16–20]. Indeed, asymptomatic colonization with *Blastocystis* spp. is associated with a higher overall diversity of intestinal bacteria [1] and with significant changes in the dominant species within the gut microbiota (e.g., a reduced relative abundance of *Prevotella copri* and an increased relative abundance of *Ruminococcus bromii*) [21].

However, colonization with *Blastocystis hominis* has also been associated with reduced levels of some intestinal immunity markers (fecal calprotectin and IgA) in humans [21]. When symptoms associated with *Blastocystis* infection are present, these are usually the result of the innate immune response following the breakdown of the intestinal barrier. There is infiltration and damage of the intestinal epithelium involving the activation of membrane receptors such as TLRs and CD8 T cells, macrophages, and neutrophil activation, including the production of immunoglobulin M (IgM), IgG, and IgA [22]. That finding suggests an anti-inflammatory environment in the intestine, likely achieved via changes in microbiome structure. Still, whether these conditions are produced specifically by *Blastocystis* or are more general responses due to colonization by other intestinal protozoa prevalent in underdeveloped areas remains unknown [23]. Many factors are likely to influence the relative pathogenicity of

intestinal microeukaryotes, including microbial shifts that expand or decrease specific classes of bacteria (taxa) that activate or regulate the immune response, the host species and its genetic background, the presence of microbe-associated molecular patterns that exist as virulence factors that induce inflammation and/or pathology, or the induction of chronic inflammation, or granulomatous reactions to contain the infection [13].

Other species of intestinal protozoa, such as *Endolimax nana*, *Entamoeba polecki*, *Entamoeba coli*, *Iodamoeba bütschlii*, and *Chilomastix mesnili*, which are not considered pathogenic, are even thought to be beneficial inhabitants of the intestine [24]. However, only a few reports on these parasites, regarding their role in the intestinal microbiota and regulation of the immune response in their host. *Blastocystis spp.*, *E. coli*, and *E. nana* have been associated with a significant increase in bacterial richness; and *E. coli* could be related to a healthy status in infants [25].

On the other hand, it has been observed that the composition of the intestinal microbiota can affect the function of tight junction proteins [26–30]. Tight junctions are dynamic and adaptable structures that serve as tissue barriers and form a protein seal that regulates the diffusion of ions and solutes between cells. This paracellular pathway plays an important role in paracellular permeability and the regulation of tissue homeostasis in critical cellular processes (e.g., cell proliferation, and migration) [26, 31]. This effect has been studied mainly in pathogenic parasites finding the tight-junction proteins can be affected directly by the presence of microbial factors [26–30] or indirectly via the activation of immune-response mechanisms related to the presence of intestinal bacteria [27, 32] and inflammation [33–36].

Because the role of non-pathogenic parasitic protozoa in the intestinal ecosystem is still poorly understood, combined efforts are needed to explore and understand the relationship of these microorganisms with the host immune response and the intestinal microbiota.

What we did know is that the interactions between the microbiota and parasites can potentially figure out the physical and immunological microenvironment of the host and could significantly influence their health status [1, 37]. In a previous study, we determined the relationship between the presence of multiple enteric parasites, such as *B. hominis*, *E. dispar*, *E. nana*, *C. mesnili*, *I. bütschlii*, *E. coli*, *Hymenolepis nana*, and *Ascaris lumbricoides* as well as the community structures of gut bacteria in an asymptomatic mother-child cohort from a semi-rural community in Mexico. We found that mothers and infants with previous parasite exposure showed significant changes in bacterial population structure, specifically in bacterial beta diversity and relative abundances of multiple bacterial taxa such as Clostridiales, Actinobacteria, and Bacteroidales [38].

In response to these findings, and to the aforementioned literature, we hypothesized that the microbiota associated with exposure to parasitic protozoa, under non-pathogenic conditions, could promote the regulation of the intestinal immune response by inducing a decrease in the inflammatory state and greater protection in the intestinal barriers, potentially determining a physical and immunological microenvironment of the host that could significantly influence its health status.

Nevertheless, knowing the effect of protozoan parasites on the immune response in humans is complicated due to ethical difficulties in performing experimental studies. Therefore, establishing a humanized gnotobiotic mice model through fecal microbiota transplantation (FMT) from human feces into germ-free mice provides an innovative and powerful tool to mimic the human microbial system [39].

Furthermore, to generate information on the immune response mediated by microbiota exposed to intestinal parasitic protists, the results of this work provide further evidence in the search for new therapeutic alternatives and the control of some forms of inflammatory bowel

disease in the human host because the administration of live microbes as a therapeutic modality is increasingly being considered [39].

In the present study, we report on the effect of FMT of asymptomatic individuals multi-parasitized with protozoa (P) in a germ-free mouse model to determine the local immune system regulation by assessing inflammatory cytokine production, T-cell proportions, colonic mucus production, gene expression of tight junction proteins, and antimicrobial peptides in germ-free mice. In addition, we used mice colonized with the microbiota not exposed to protozoa as a control group (NP).

## Materials and methods

### Study design and characteristics of the studied participants

The participants are part of a cross-sectional cohort of mother-child binomials studied in previous research [38]. The recruitment of this cohort of mother-child binomials began from March 3, 2015, to March 29, 2016, in a semi-rural Mexican population of Xoxocotla, state of Morelos, 250 km south of Mexico City, with high levels of exposure to intestinal parasites. This study was conducted in collaboration with Morelos Health Services.

In this work, we selected eight mother-child binomials: four corresponding to the P, and four to the NP groups. All mother-child binomials donated fresh fecal samples. However, in the case of two children (one in the P, and one in the NP group), there were not enough fecal samples to include them in this study.

The P group (n = 7) consisted of mothers (ID: P1-P4; n = 4), between 18 and 31 years old, with an average age of 25.75 years (±5.56); only 50% of the mothers have a basic level of education, 25% have soil floor, 75% have piped water service, 100% have a septic tank for drainage and 75% have formal electricity service. As for female children (ID: P5-P7; n = 3), born by vaginal delivery, the age range is between 20 and 23 months, with an average age of 22 months (±1.73); 100% were breastfed and began breastfeeding between six and seven months of age. All volunteers in this group had a protozoan multi-parasitosis during sampling but did not present gastrointestinal symptoms. The parasites present in the parasite group were: *B. hominis* in 100% of the volunteers, *E. coli* in 57.14%, *E. dispar* in 57.14%, *E. nana* in 42.86%, *I. bütschlii* in 28.57%, and *Chilomastix mesnili* in 14.29% of volunteers (S1 Fig).

The NP group (n = 7) consisted of mothers (ID: NP1-NP7; n = 4), between 20 and 32 years old, with an average age of 32.75 years (± 1.89). 50% of mothers have a technical degree and the other 50% have high school. The marital status of 75% of the mothers is married. 25% of the mothers have a soil floor, 25% do not have drinking water and electricity service and 100% have a septic tank for drainage. As for female children (ID: np5-np-7; n = 3), born by vaginal delivery, the age range is between 9 and 15 months, with an average age of 12 months (± 3); 100% were breastfed and began breastfeeding between five and eight months of age. All the participants (n = 14) reported no gastrointestinal symptoms attributable to parasitic infection and had not used antibiotics or other drugs for at least six months before the recruitment (S1 Fig). In both, P and NP groups, the most abundant orders were Bacteroidales, Clostridiales, and Bifidobacteriales. However, the orders Bacteroidales, Coriobacteriales, Aeromonadales, Fusobacteriales, Actinomyacetales, and Verrucomicrobiales are more abundant in NP group, while the orders Clostridiales, Bifidobacteriales, and Campylobacteriales are higher in P group (S2 Fig).

### Ethical considerations

Every volunteer mother was informed about the details of the project, the objectives, and the advantages of participation, along with the biological samples needed, the sampling

procedures, and possible complications that could arise. Participating mothers signed a written informed consent letter for their children and themselves, before sample collection. The information of participants was coded for registration in the database, maintaining the confidentiality of each participant.

The research protocol was approved by the Ethical and Scientific Committee of the Faculty of Medicine of the Universidad Nacional Autónoma de México (FM/DI/006/2014); it was further informed by the Official Norm NOM-012-SSA3-2012 for Research in Humans from the Mexican Health Ministry and was conducted by the Declaration of Helsinki, and NOM-062-ZOO-1999 about the use of experimental animals in research projects. All animal procedures were reviewed and approved by the Animal Care Committee (ACC) of the University of British Columbia (Anim. Care Cert. #: A13-0265) and followed the standard operating procedures and policies of the Canadian Council on Animal Care guidelines. All the procedures were made to minimize suffering with the support of qualified personnel for each procedure.

## Fecal microbiota transplantation (FMT) procedure

To measure the effects of FMT-P on the regulation of the immune response, we created a humanized murine model using FMT by repeated oral gavage of germ-free female Swiss-Webster mice. Fecal samples were grouped as follows: 1) P: fecal samples of participants diagnosed as parasitized with more than one intestinal protozoan, and 2) NP: fecal samples of participants without parasites diagnosed. Each group consisted of fecal samples of four female adults (mothers) and three female children (daughters).

The humanized murine FMT model consisted of microbiota suspensions, from parasite and non-parasite fecal samples, prepared for oral transplantation into germ-free mice [40].

Briefly, all fecal samples were frozen at −80°C. Later, 50 mg of each fecal sample was thawed on ice and dissolved in 1 mL of reduced sterile PBS (0.05% w/v of cysteine-monohydrate-monohydrochloride). With the purpose that possible detritus particles, protozoa cysts, or fungi forms could be separated from the fecal suspension, this was centrifuged in 1 mil vials at 1200 g for 3 minutes, where, mainly, the bacteria were in the supernatant [41]. The supernatant was transferred to a transportation bubble in anaerobic conditions (90% N2, 5% CO2, 5% H2). Three-to-four-week-old germ-free Swiss Webster female mice (Taconic Biosciences, Cambridge City, IN, USA) were transplanted with 0.1 mL of human donor suspension via oral gavage. We used a syringe with a gavage needle and deposited the suspension directly into the animals' stomachs. The procedure was repeated three more times at 48-hour intervals. Between four and five mice were transplanted with the same material from each parasitized or non-parasitized fecal sample (The number of replicates was conditioned by the volume quantity of fecal samples: 5 replicates for P1-P4, 5 for NP1-NP4, 4 for P5-P7, and 4 for NP5-NP7; n = 64 transplanted mice).

All mice were housed in a controlled environment with 12 h light-dark cycles (07.00 h–19.00 h light; 19.00 h–07:00 h dark) at a constant temperature (24.0 ± 0.2°C). The animals were given adequate access to food and water. All mice survived the treatment before the euthanization date.

Three weeks after the last gavage, the mice were euthanized using 5% isoflurane anesthesia, followed by carbon dioxide exposure (2–3 minutes), and to ensure the death of the animals, cervical dislocation was performed. Intestine (duodenum, jejunum, ileum, and colon), spleen, and liver samples were collected from each transplanted mouse and preserved for later experiments.

## Measurements of inflammatory cytokine production

Tissue samples were harvested in ice-cold PBS. The fecal matter was removed, and the intestines were opened along their axis and thoroughly washed with ice-cold PBS. Next, intestinal, spleen, and liver samples were cut into small pieces and incubated in 500 μL of complete RPMI media (GIBCO, Grand Island, NY, USA) (10% HIFBS, 2 mM glutamine (1X), 100 U/mL penicillin, 100 μg/mL streptomycin, and 25 mM HEPES) with protease inhibitors (one tablet of Complete Mini, EDTA-free Protease inhibitor cocktail (Roche, Basel, Switzerland) per 10 mL of solution) at 37.0°C in 48 well plates. After 48 hours of incubation, the supernatants were removed from the wells and stored at -80.0 C, and the dry tissue weight was recorded. The cytokines IL-6, TNF, MCP-1, IFN-γ, IL-12, and IL-10 were quantified utilizing Cytometric Bead Array assays (BD Biosciences, US) following the manufacturer's instructions. Briefly, the FACSCalibur (BD Biosciences) flow cytometer was configured and calibrated according to the kit's instructions and specifications. A concentration curve was obtained by duplicating the fresh kit's standards for every analyte. The sample supernatants were tested as described in the kit's instructions, and their concentration was determined by interpolating the median fluorescence intensity of each sample measured using the concentration curve and FlowJo software (BD Biosciences). We next used logistic regression analysis built into Graph-Pad Prism software as a model to adjust the curves for every cytokine. The concentration values were finally normalized to the weight of the tissue (pg/mL).

## Lymphocyte isolation

Immediately after the mice were euthanized, their colon and spleen tissue samples were placed in complete RPMI 1640 media (2 mM L-Glutamine and 10% fetal bovine serum (FBS)) (GIBCO), and the tissues were stored on ice for transportation and processing. To isolate the lymphocytes from the lamina propria, the colon samples were cut open to remove the fat and Peyer's patches and then washed with cold PBS to remove fecal matter and mucus before being placed in complete RPMI on ice. To remove the intestinal epithelial cells, the colon samples were washed with warm strip buffer (PBS+/+ (with Ca and Mg), 5% FBS, 1mM dithiothreitol (DTT), and 1 mM EDTA) for 30 minutes at 37.0 C in agitation. Finally, the supernatant was discarded. The tissue was washed twice with cold RPMI (10% FBS) and incubated with digest buffer (RPMI, Collagenase/Dispase 0.5 mg/mL, and DNaseI 0.02 mg/ml) to isolate the lamina propria lymphocytes (LPLs). After being shaken for 30 minutes at 37.0 C, the tissue was lightly mashed, and the supernatant was filtered with a 70-μm filter to isolate the LPLs and then resuspended with RPMI (10% FBS).

To isolate the spleen lymphocytes, the samples were placed in Petri dishes with 5 mL of Hank's balanced salt solution (HBSS) buffer (GIBCO), and were carefully cut into small pieces measuring roughly 0.2 centimeters on a side. Those pieces were then incubated for 20 minutes at 37.0°C with 5 mL of HBSS containing Collagenase IV (GIBCO) (100 U/mL) and DNase solution (Thermo Scientific) (20 μg/mL) with 1% FBS (GIBCO). EDTA 1 mM/mL (UltraPure) was added, and the resulting solution was incubated for 5 minutes at room temperature to stop the enzymatic reaction. Using the plunger end of a syringe, the excised spleen was mashed through a strainer, and 10 mL of cold PBS (Gibco) was added to wash the cells through the strainer. The cells were centrifuged at 400–600g for 5 minutes at 4.0°C, and the supernatant was discarded. The resulting pellet was resuspended in 5 mL of cold 1x RBC Lysis buffer (eBioscience, San Diego, CA, USA) and incubated for 5 minutes on ice. Fifteen mL of cold PBS (Gibco) was then added and centrifuged at 400–600g for 5 minutes at 4.0°C, and the supernatant was discarded. The pellet was then resuspended with RPMI (10% FBS).

## Lymphocyte purification and flow cytometry

The lymphocytes were purified in a 40% Percoll gradient and resuspended in RPMI (5% FBS). They were counted using an automatic cell counter (Countess, BD Biosciences), and the cells were stained with a 1/200 dilution of fluorochrome-conjugated antibodies against CD45 (clone 30F-11), CD3ε (eBio500A2), CD49b (clone DX5), CD8 (53–6.7), CD4 (RM4–5), and Foxp3 (FJK-16S) (eBioscience). We analyzed their populations using an LSR II flow cytometer (BD Biosciences) with the CellQuest and FlowJo (version 8.7) software packages. We measured the relative frequencies of the CD45+, CD3+, CD49b+, CD8+, CD4+, and Foxp3+ cell populations in the colon lamina propria and the spleens of the mice using flow cytometry.

## Colonic mucus width determination

To obtain an intact layer of mucosa and mucus, colon sections measuring 1 centimeter in length containing a fecal pellet were collected from the mice and placed immediately into fresh methanol-Carnoy's fixative for 2 hours at 4.0˚C and then transferred into 100% ethanol at 4.0˚C. The paraffin-embedded tissues were next cut into 5-μm slices and stained with Alcian Blue-periodic acid Schiff to enable measurements of the thickness of their mucus layer via microscopy.

## RNA extraction, cDNA conversion, and relative gene expression RT-qPCR assays

To preserve the RNA integrity from the animals' intestinal samples were immediately kept in RNA later (Qiagen, Hilden, Germany) on ice for transportation and placed in a freezer overnight at -70.0 C for storage. The RNA extraction was conducted using up to 30 mg of tissue and a RNeasy Mini Kit (Qiagen). Briefly, the tissue was disrupted in RNAase-free tubes with β-mercaptoethanol and Buffer RLT from the RNeasy Mini Kit. A flamed and cool tungsten bead was added to each tube, and the tissue was homogenized using a Mixer Mill (Retsch/Qiagen) at a frequency of 1 in 25 for 2 minutes. The RNA was precipitated and isolated, and the DNase digestions were conducted according to the manufacturer's instructions. We assessed RNA concentration and purity with a NanoDrop 1000 (Thermo Fischer Scientific), and cDNA was synthesized using the QuantiTect Reverse Transcription kit (Qiagen) with 1 μg of RNA as a template.

We evaluated gene expression with RT-qPCR using an Applied Biosystems 7500 machine with a QuantiTect SYBR green master mix (Qiagen) with 10 μL of reaction volume containing 8 pmol of each primer. We used the following primer sequences: muc2 (Fwd- 5′-GCTGACG AGTGGTTGGTGAATG-3′, Rev- 5′-GATGAGGTGGCAGACAGGAGAC-3′), tjp1 (Fwd- 5′-CCCTGAAAGAAGCGATTCAG-3′, Rev- 5′-CCCGCCTTCTGTATCTGTGT-3′), cldn2 (Fwd-5′-ATACTACCCTTTAGCCCTGACCGAGA-3′, Rev- 5′-CAGTAGGAGCACACATAACAGC TACCAC-3′), ocln (Fwd- 5′-ACTGGGTCAGGGAATATCCA-3′, Rev- 5′-TCAGCAGCA GCCATGTACTC-3′), reg3γ (Fwd- 5′-AAGCTTCCTTCCTGTCCTCC-3′, Rev- 5′-TCC ACCTCTGTTGGGTTCAT-3′), and relmβ (Fwd- 5′-GCTCTTCCCTTTCCTTCTCCAA-3′, Rev- 5′-AACACAGTGTAGGCTTCATGCTGTA-3′). The gene encoding glyceraldehyde phosphate dehydrogenase (gapdh) was used as an endogenous control for normalization (Fwd- 5′-ATTGTCAGCAATGCATCCTG-3′, Rev- 5′ATGGACTGTGGTCATGAGCC-3′). The amplification conditions were as follows: 95.0˚C for 10 minutes and 40 cycles of 95.0˚C for 15 seconds, 60.0˚C for 1 minute, and a melt curve of 95.0˚C for 15 seconds, 60.0˚C for 1 minute, 95.0˚C for 30 seconds and 60.0˚C for 15 seconds. We calculated relative gene expression using the ΔΔC(t) method relative to the control mice.

## Statistical analysis

Our results are expressed as means and standard deviations (SDs), unless otherwise specified. The statistical significance of the differences between the experimental group (germ-free mice transplanted with microbiota from parasitized individuals) and the control group (germ-free mice transplanted with microbiota from non-parasitized individuals) was calculated using a two-tailed Student's t-test or the Mann-Whitney U-test (for non-parametric data). The statistical analyses were performed using GraphPad Prism Software (version 9.00) (GraphPad Software, www.graphpad.com).

## Results

### Gut colonization of germ-free mice with FMT-P suppresses local inflammatory cytokine production

We used flow cytometry to quantify the production of the cytokines interleukin (IL)-6, tumor necrosis factor (TNF), interferon-gamma (IFN-γ), monocyte chemoattractant protein-1 (MCP-1), IL-12, and IL-10 in the intestines, spleens, and livers of the mice receiving FMT treatment. In general, mice with FMT-P exhibited lower levels of most inflammatory cytokines in their small intestine compared with mice with FMT-NP (Fig 1). In the duodenum, we observed significantly lower production of IL-6, TNF, and IFN-γ in mice with FBT-P compared with mice with FMT-NP (Fig 1A). In addition, IL-6, TNF, MCP-1, and IL-12 production decreased significantly in the jejunum (Fig 1B).

Furthermore, we observed a remarkable decrease in IL-10 levels in mice FMT-P. Nevertheless, differences in the levels of all the tested cytokines in the ileum or colon in both tested groups were not statistically significant (S3 Fig). The only statistically significant difference in cytokine production was noted in the spleens and livers of mice FMT-P or FMT-NP. We observed an increase in TNF levels in mice with FMT-P (Fig 2).

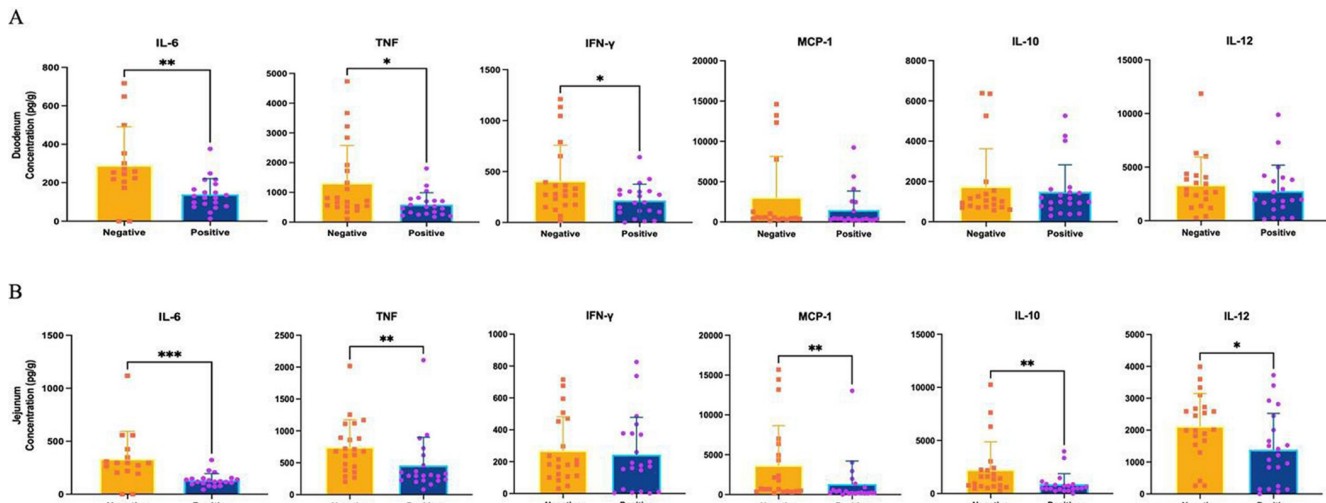

**Fig 1. Cytokine production in mice transplanted with gut bacteriome material from individuals parasitized with intestinal protozoa.** Inflammatory cytokine concentration (pg/g) in the duodenum (A) and jejunum (B) of mice that received protist-associated and protist-negative FMT. Cytokine concentration was measured using cytometric bead array assays in the intestine of germ-free mice that received protist-negative and protist-associated FMT. The results are the means of three experiments and are presented with their standard deviation; differences were tested using the Mann-Whitney U-test. The symbols *, **, *** correspond to different levels of statistical significance.

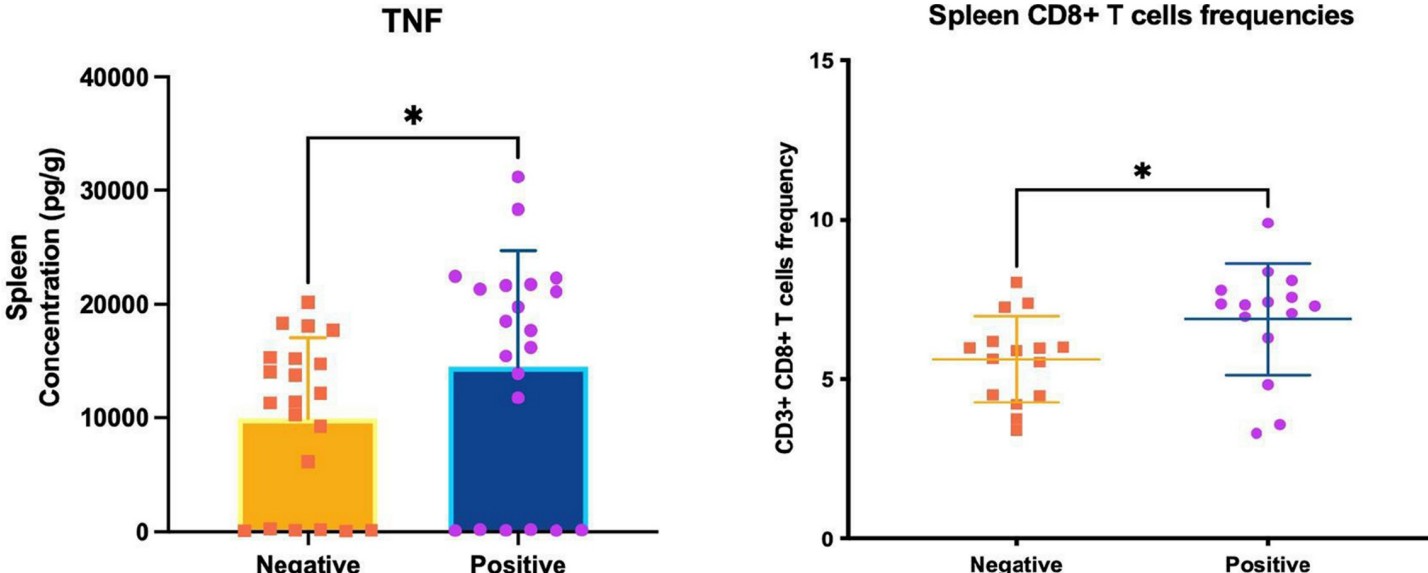

**Fig 2. TNF production and the frequency of CD8+ T cells in the spleens of mice transplanted with feces from individuals with intestinal protists.** (A) TNF concentration (pg/g) was measured via flow cytometry using cytometric bead array assays in the spleens of germ-free mice that received protist-negative and protist-associated FMT. (B) CD3+ CD8+ T cell frequencies (%) in the spleens of germ-free mice that received protist-negative and protist-associated FMT; data measured by flow cytometry. The results are the means of three experiments and are presented with their standard deviation; differences were tested using the Mann-Whitney U-test. * corresponds to statistical significance.

## Gut colonization of germ-free mice with fecal microbiota previously exposed to protozoa is associated with a smaller fraction of Treg cells in the colon and spleen

We determined the proportions of T lymphocytes in the colons and spleens of mice with FMT using flow cytometry. Our goal was to assess whether a protozoa-associated microbiota can alter immune cell populations. Our results revealed non-statistically significant differences in NK cell frequencies in both sampled organs (S4 Fig). We observed a higher frequency of CD8 + T cells in the spleens of mice colonized with FMT-P, compared with mice with FMT-NP (Fig 2).

When we examined CD4+ T cell populations, we found a significant reduction in the frequencies of CD3+/CD4+/Foxp3+regulatory T (Treg) cells after FMT-P. That result persisted in both the colonic immune cell population as well as in the splenic immune cells compared with the FMT-NP (Fig 3).

## Gut colonization of germ-free mice with FMT-P induced a thicker colonic mucus layer

To determine whether FMT-P modified gut mucus production in transplanted germ-free mice, we measured the thickness of the colonic mucus layer using Alcian blue PAS staining (Fig 4). We measured an increase in colonic mucus layer thickness in mice colonized with protozoa-associated microbiota material compared with mice colonized with microbiota from non-parasitized individuals with protozoa-negative feces (Fig 4A). To further investigate this phenotypic difference, we analyzed the expression of muc2, the gene encoding the mucin glycoprotein MUC2, using RT-qPCR. We found a statistically significant increase in the expression muc2 in the colon of mice colonized with FMT-P in contrast with the levels observed in mice colonized with FMT-NP (Fig 4B).

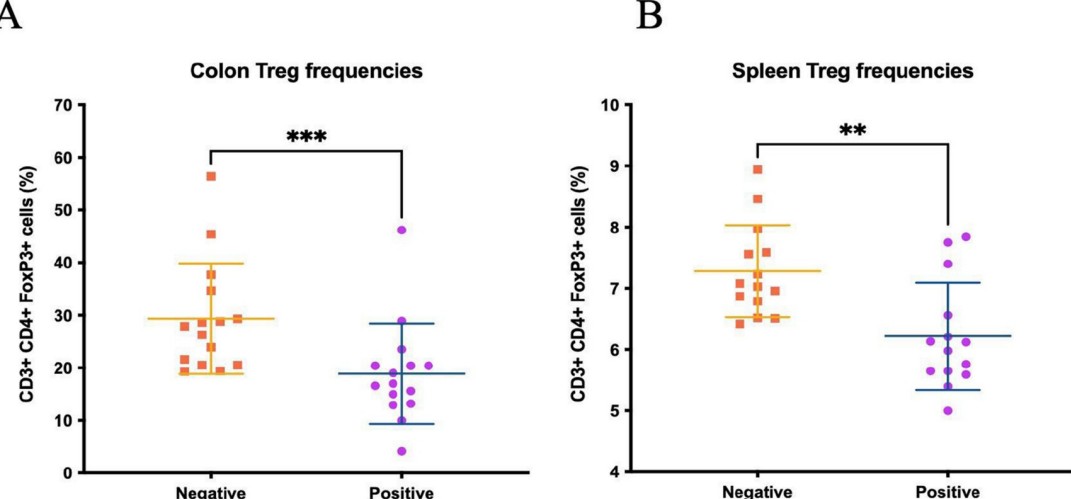

**Fig 3. Frequencies of regulatory T cell in mice transplanted with feces from donors with intestinal protists.** Flow cytometry frequencies (%) of CD3+ CD4+ Foxp3+ Treg cells in the colons (A) and spleens (B) of germ-free mice that received protist-negative and protist-associated FMT. The results are the means of three experiments and are presented with their standard deviation; differences were tested using the Mann-Whitney U-test. **, *** correspond to different levels of statistical significance.

## Differences in the gene expression of markers of epithelial integrity in germ-free mice with FMT-P

We used RT-qPCR to measure relative gene expression to determine the effect of FMT on the integrity of the intestinal epithelium in mice colonized with the microbiota from parasitized individuals compared with the FMT-NP mice group (Fig 5). We focus on analyzing genes that

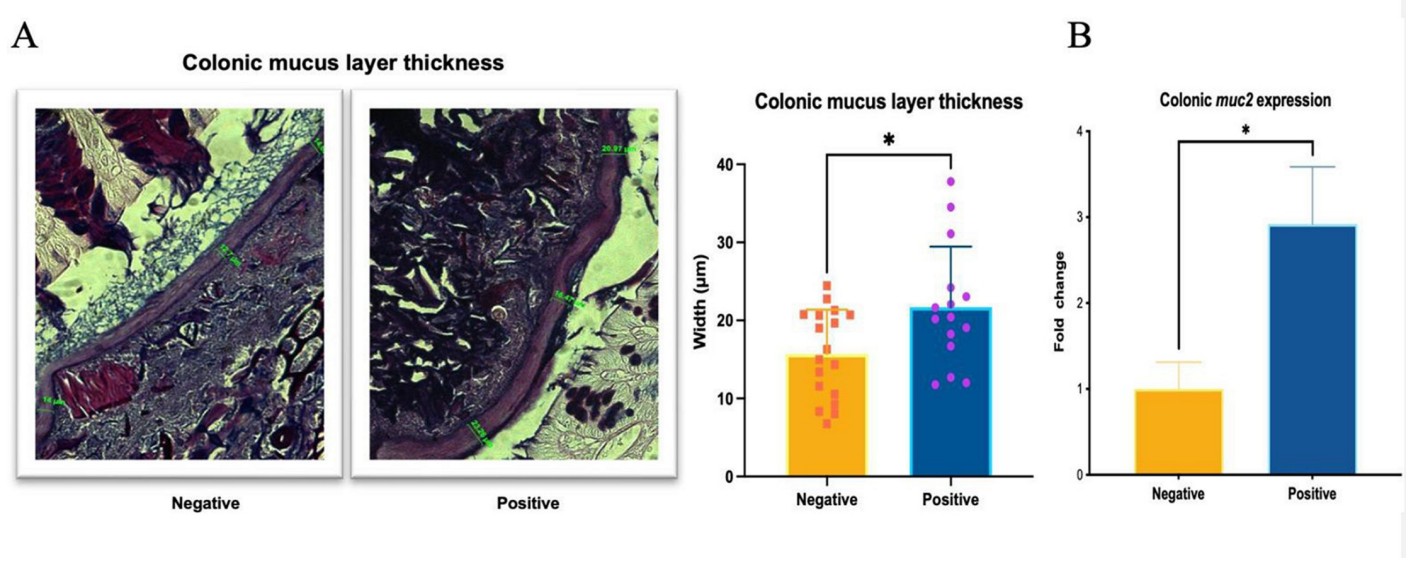

**Fig 4. The colonic mucus layer in the intestines of mice transplanted with feces from individuals with intestinal protists.** (A) Colonic mucus layer width (μm) in germ-free mice that received protist-negative and protist-associated FMT; data measured via Alcian Blue-PAS staining histology. (B) Relative gene expression (fold change) of muc2 in germ-free mice that received protist-negative (n = 15) and protist-associated (n = 15) FMT; data measured by RT-qPCR. The results are the means of three experiments and are presented with their standard deviation for the colonic mucus layer width and mean (SEM) for muc2 expression. Differences were tested using the Mann-Whitney U-test. **, *** correspond to different levels of statistical significance.

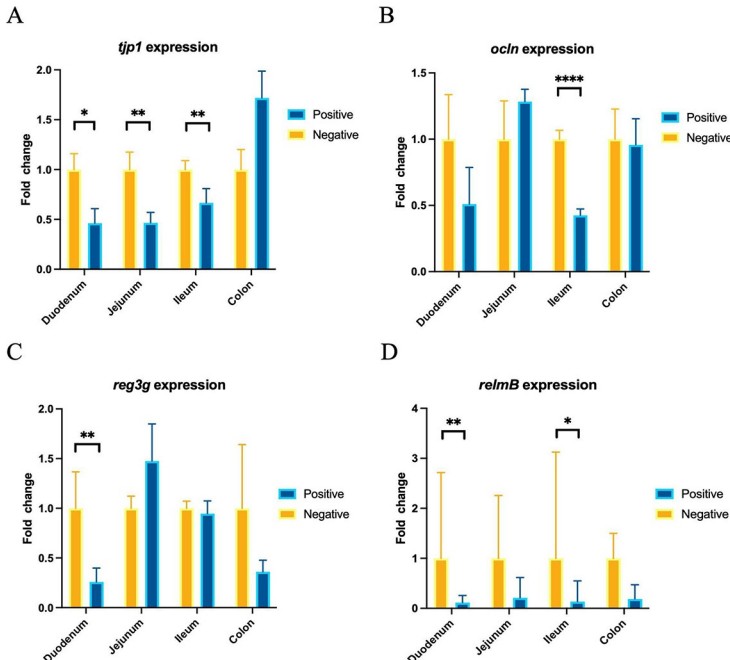

**Fig 5. Relative gene expression of epithelial integrity markers in the intestines of mice transplanted with human feces containing intestinal protists.** Relative gene expression (fold change), measured by RT-qPCR, of the tight junction proteins tjp1 (A) and ocln (B). The AMPs reg3γ (C) and relmβ (D) in the intestines of germ-free mice that received protist-negative (n = 15) and protist-associated (n = 15) FMT. The results are the means of three experiments and are presented with their SEM. Differences were tested using the Mann-Whitney U-test. *, **, **** correspond to different levels of statistical significance.

encode epithelial proteins, mainly at the colon level, because it is the target organ of the parasites. The expression level of tjp1, ocln, reg3γ, and relmβ in the colon of mice with FMT-P was not significantly different from that of the FMT-NP mice group. However, some interesting trends were observed in the increase of the expression of tjp1 in mice with FMT-P in contrast with the FMT-NP mice group. We found that the expression of tjp1 in mice with FMT-P was decreased across all three regions of the small intestine compared with the levels observed in mice with FMT-NP (Fig 5A). Furthermore, we observed a significant decrease in ocln, and relmβ expression in the ileum (Fig 5B and 5D) and a decrease of reg3γ as well as relmβ in the duodenum of mice with FMT-P (Fig 5C and 5D).

## Discussion

It is known that interactions between microbiota and parasites can potentially determine the physical and immunological microenvironment of the host and could significantly influence its health status [1]. In this work, we analyzed the regulation of the immune response of germ-free mice transplanted with FMT-P in contrast with the FMT-NP mice group.

It is important to note that our results should be interpreted with caution. The sample size is a limitation of our work, and our results cannot be generalized. They also do not allow us to analyze the response induced by different associations or consortia of protozoa found in the samples. Thus, further studies with an increased sample size are necessary for a better understanding. Another limitation is that in this work only the indirect influence of the analyzed parasites will be observed, since the model took into account a freeze/thaw cycle, which increasingly decreased the parasite populations or inactivated them, so in future research, it

will be necessary to analyze the immune response taking into account fresh fecal samples and thus generate greater knowledge about the direct influence of the presence of these microorganisms on the microbiome and the immune system in fecal transplant models in mice, as well as analyze their physiological and clinical status, which will allow us to generate more forceful conclusions.

Despite these limitations, this work provides important information on the immune response of intestinal parasitic protozoa from a fecal microbiota transplant model. In addition, this is pioneering research on this type of protozoa on the immune system in an FMT mouse model.

Here we hypothesized that the microbiota associated with exposure to parasitic protozoa, under non-pathogenic conditions, could promote the regulation of the intestinal immune response by inducing a decrease in the inflammatory state and greater protection in the intestinal barriers, potentially determining a physical and immunological microenvironment of the host that could significantly influence its health status. Nevertheless, our hypothesis was only partially fulfilled.

Indeed, our findings revealed a decrease in the inflammatory state of the gut mediated by fecal microbiota exposed to gut protozoa. Here we found that decreased the production of the proinflammatory cytokines IL-6, TNF, MCP-1, IL-12, and IFN-γ in the small intestine of germ-free mice with FMT-P. This agrees with other studies indicating that *B. hominis*, can also downregulate the host inflammatory response conditions [21, 22]. Also, our group, previously reported that *Blastocystis* spp. downregulates the inflammatory response in the human intestine [23]. We also detected lower levels of fecal calprotectin, an important marker of neutrophil and other myeloid cell activity [42–45], and lower IgA production in people colonized by this protozoa [21]. Similarly, *E. histolytica* promotes the downregulation of IL-5, IL-6, TNF, and IFN-γ in nonpathogenic conditions [46]. We also observed a significant decrease in IL-10 production in the jejunum of mice transplanted with microbiota from protozoan-infected donors. In contrast, the intestinal microbiota of helminth-infected mice increases the levels of IL-10 [47, 48]. We speculated that, in a non-pathogenic environment, microbiota exposed to parasitic protozoa might have less aggressive interactions with the intestinal epithelium, which could lead to a downregulation of the inflammatory response in the intestine. This anti-inflammatory effect supports the "old friends" hypothesis, which is based on the ability of parasitism to provoke immunomodulatory effects on the host immune system; such a process has important implications for preventing autoimmune and inflammatory disorders [49]. In contrast, a proinflammatory effect is associated with colonization of pathogenic parasites in the gut [50, 51].

Intestinal Treg cells are principally present in the colon, suppressing inflammatory responses by producing anti-inflammatory cytokines (e.g., IL-10 and TGF-β). But unlike the elevated numbers of Foxp3-expressing regulatory T cells reported with helminth infections [52], we observed fewer CD4+/Foxp3+ Treg cells in the colons of mice with FMT-P.

Furthermore, we noted elevated TNF production and CD3+/CD8+ T cell frequencies in the spleens and livers of mice with FMT-P. On the other hand, mice with FMT-NP exhibited lower levels of TNF production and CD3+/CD8+ T cells. That finding suggests a systemic inflammatory activation mediated by previously exposed to protozoa microbiota. T cells are important sources of TNF, but, at the same time, they are also targets of TNF activity. TNF can have an anti-inflammatory effect in specific settings of CD8+ T cell activity (e.g., apoptosis induction after a viral infection) [53]. Moreover, TNF possesses a strong pro-inflammatory effect characterized by CD8+ T cell proliferation, T cell coactivation, and boosted T cell survival during the early stages of inflammatory responses [54, 55].

We also found a lower expression of tjp1 in the duodenum, jejunum, and ileum of germ-free mice with FMT-P. The tjp1 has been linked to the assembly of functional junctions and

signal transduction in the epithelium [56, 57]. We also observed a lower gene expression of ocln in the ileum of mice with FMT-P. Occluding is required for cytokine-induced regulation of the tight junction paracellular permeability barrier [36, 58, 59]. It is known that changes in the expression of these different tight junction proteins alter the resistance of the intestinal barrier Nevertheless, these results could also be due to the combined effect of the transplant with fecal microbiota exposed to intestinal parasites and the mouse strain used. In mice, it has been observed that the genetic background modulates the intestinal barrier, which varies according to the mice strain [60].

Despite what has been said for the small intestine, when we analyze what happens in the colon in mice, the target organ for parasites, we observe a greater expression tendency of tjp1. However, we did not find statistical significance in these results, hence future studies with a larger sample size will be necessary to confirm these results. Likewise, additional epithelial permeability experiments are required to determine whether the reduced gene expression of tight junction-associated proteins observed in our model in the duodenum and jejunum is mediated by the microbiota profile of parasitized individuals, or the mouse strains or by the combined effect of both transplantation with fecal microbiota exposed to intestinal parasites and the mouse strain used and thus also to determine the consequences on health status.

Furthermore, we determined the gene expression of the AMPs reg3γ and relmβ. These AMPs are important in bacterial growth regulation and placement to maintain intestinal homeostasis [61]. Interestingly, we noted a significantly lower expression of reg3γ in the duodenum, and relmβ in the duodenum and the ileum of mice with FMT-P compared with mice with FMT_NP. It is known that the levels of AMP are lower in the guts of protozoa-infected individuals [62, 63]. One of the primary inducers of reg3γ expression is the short-chain fatty acid butyrate, produced by bacteria of the Firmicutes phylum. Members of the families Ruminococcaceae and Clostridiaceae are found in the human colon [64]. Even though the microbiota from protozoa-exposed individuals was rich in Clostridia, mice that received the transplantations exhibited lower expression of AMPs. The decreased direct contact between intestinal bacteria and the gut epithelium is one possible explanation for this result. Furthermore, this finding might also suggest that the lower AMP expression is achieved through other mechanisms, in which the intestinal mucus layer could play a regulatory role.

We found a significant thickening of the colonic mucus layer in mice transplanted with FMT-P. In line with our findings, some reports mention that protozoa and intestinal bacteria can show similar effects on the mucosal gut. Some reports emphasize the fact that the presence of a larger mucus layer is strongly related to the intestinal microbiota, impairing the translocation of intestinal microorganisms [62, 63, 65].

Also, because the mucus layer functions as a mesh that retains bactericidal proteins, such as reg3γ and relmβ AMPs, our results suggest that the increase in the mucus layer of parasitized individuals decreases the expression of AMPs. However, further studies are needed to assess the implications of the high mucus production in mice transplanted with fecal matter exposed to parasites on the mechanisms regulating the expression of molecules involved in intestinal permeability.

## Conclusions and perspectives

This study is the first investigation designed to characterize the role of the gut microbiota determined by non-pathogenic gut protozoa and using FMT from individuals exposed to protozoa in a germ-free mouse model. Our results reinforce previous findings supporting the modulatory role of gut parasites.

We suggest that gut protozoa, such as *B. hominis*, *E. coli*, *E. nana*, *E. histolytica/E. dispar*, *I. bütschlii*, and *Chilomastix mesnili*, under multi-parasitic conditions, affect the immunoinflammatory state in the gut via the gut microbiota. Our work also provides new insights into the biological complexity and suggests that protozoa might play a regulatory and possibly protective role in maintaining gut homeostasis in a host.

However, it remains unknown whether host immune regulation through an interaction of parasitic protozoa and bacteria plays a significant role in resistance to pathogen infection or susceptibility to host disease. Further studies are needed to analyze the direct relationship of the studied intestinal parasites with the associated intestinal microbiota on the immune response and its implications on host health using FMT in the germ-free mouse model.

## Supporting information

**S1 Fig. Data and percentages of parasitic protists in the group of volunteers studied.**
(PDF)

**S2 Fig. The most abundant orders of bacteria in the parasitized and non-parasitized groups of volunteers.**
(PDF)

**S3 Fig. Cytokine production in mice transplanted with feces from people with intestinal protists.** Inflammatory cytokine concentration (pg/g) measured by cytometric bead array assays in the intestine of germ-free mice with protist-negative and protist-associated FMT. Inflammatory cytokines concentrations in the ileum (**A**), colon (**B**), spleen (**C**), and liver (**D**) of mice with protist-associated and protist-negative FMT are shown as a pool of three experiments and are represented as mean (SD) and differences were tested with Mann–Whitney U-test. p = probability value.
(TIF)

**S4 Fig. NK cell frequencies in the colon and spleen of mice with FMT from people with intestinal protists.** Flow cytometry frequencies (%) of CD3[+] CD49b [+] NK cells in the colon (**A**) and the spleen (**B**) of germ-free mice with protist-negative and protist-associated FMT. Results are a pool of three experiments and are represented as mean (SD) and differences were tested with the Mann–Whitney U-test. p = probability value.
(TIF)

## Acknowledgments

The authors would like to thank the PhD Miriam E. Nieves-Ramirez, Alvaro de la Mora-Ramiro, M.Sc. Martha Zaragoza-Martinez and the chemist Angeles Padilla-Mendoza for their invaluable technical support, as well as Mrs. Margarita García for her participation in the processing of fecal samples. We want to thank all the volunteers who participated in this study. Finally, we thank to reviewers for their helpful comments and suggestions to improve the manuscript.

## Author Contributions

**Conceptualization:** Oswaldo Partida-Rodríguez, Eric M. Brown, Marie-Claire Arrieta, Javier Torres, Cecilia Ximénez, B. B. Finlay.

**Data curation:** Oswaldo Partida-Rodríguez, Eric M. Brown, Mihai Cirstea, Lisa A. Reynolds, Charisse Petersen, Eric G. Hernández, Liliana Rojas-Velázquez, Patricia Moran, Enrique González Rivas, Angélica Serrano-Vázquez, Horacio Pérez-Juárez.

**Formal analysis:** Oswaldo Partida-Rodríguez, Eric M. Brown, Mihai Cirstea, Lisa A. Reynolds, Charisse Petersen, Jorge Peña-Díaz.

**Funding acquisition:** Oswaldo Partida-Rodríguez, Javier Torres, B. B. Finlay.

**Investigation:** Oswaldo Partida-Rodríguez, Eric M. Brown, Sarah E. Woodward, Lisa A. Reynolds, Jorge Peña-Díaz, Marie-Claire Arrieta, B. B. Finlay.

**Methodology:** Oswaldo Partida-Rodríguez, Eric M. Brown, Sarah E. Woodward, Mihai Cirstea, Lisa A. Reynolds, Charisse Petersen, Stefanie L. Vogt, Jorge Peña-Díaz, Lisa Thorson, Marie-Claire Arrieta, Eric G. Hernández, Patricia Moran, Enrique González Rivas, Angélica Serrano-Vázquez, Horacio Pérez-Juárez, Javier Torres, B. B. Finlay.

**Project administration:** Oswaldo Partida-Rodríguez, Stefanie L. Vogt, Marie-Claire Arrieta, Javier Torres, Cecilia Ximénez, B. B. Finlay.

**Resources:** Lisa Thorson, Eric G. Hernández, Liliana Rojas-Velázquez, Javier Torres, B. B. Finlay.

**Software:** Oswaldo Partida-Rodríguez, Eric M. Brown, Sarah E. Woodward, Mihai Cirstea, Lisa A. Reynolds, Charisse Petersen.

**Supervision:** Oswaldo Partida-Rodríguez, Eric M. Brown, Sarah E. Woodward, Mihai Cirstea, Lisa A. Reynolds, Charisse Petersen, Stefanie L. Vogt, Jorge Peña-Díaz, Lisa Thorson, Cecilia Ximénez, B. B. Finlay.

**Validation:** Oswaldo Partida-Rodríguez, Eric M. Brown, Mihai Cirstea, Lisa A. Reynolds, Charisse Petersen, Jorge Peña-Díaz, Javier Torres, Cecilia Ximénez, B. B. Finlay.

**Visualization:** Mihai Cirstea.

**Writing – original draft:** Oswaldo Partida-Rodríguez.

**Writing – review & editing:** Eric M. Brown, Mihai Cirstea, Lisa A. Reynolds, Marie-Claire Arrieta, B. B. Finlay.

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
