## [Decision Letter · Decision Letter 0]

15 May 2024

PONE-D-24-07909Fecal bacteriome transplantation from protozoa-exposed donors downregulates immune response in a germ-free mouse model, its role in immune response and physiology of the intestine.PLOS ONE

Dear Dr. Ximenez,

Thank you for submitting your manuscript to PLOS ONE. After careful consideration, we feel that it has merit but does not fully meet PLOS ONE’s publication criteria as it currently stands. Therefore, we invite you to submit a revised version of the manuscript that addresses the points raised during the review process.

We look forward to receiving your revised manuscript.

Kind regards,

Brenda A Wilson, Ph.D.

Academic Editor

PLOS ONE

Journal Requirements:

"PAPIIT program at the National Autonomous University of Mexico (grant numbers IN226511, IN218214, and IN217821), IMSS (grant FIS/IMSS/PROT/1368). National Council of Sciences and Technology in Mexico (CONACyT; grants numbers 140990, 272601, 283522, and 257091) to C.X. and J.T., and the Canadian Institutes for Health Research to B.B.F. The funders had no role in study design, data collection and interpretation, or the decision to submit the work for publication. O.P.-R. (proposal number 208253) received a one-year scholarship from the “Estancias Posdoctorales en el Extranjero para la Consolidación de Grupos de Investigación” program of CONACyT."

4. Please expand the acronym “CONACyT” and "IMSS" (as indicated in your financial disclosure) so that it states the name of your funders in full.

**Additional Editor Comments:**

Both of the reviewers were positive but noted a number of points that need to be adequately addressed in a majorly revised manuscript before further consideration can be made.

Reviewers' comments:

Reviewer's Responses to Questions

**Comments to the Author**

1. Is the manuscript technically sound, and do the data support the conclusions?

Reviewer #1: Partly

Reviewer #2: Yes

2. Has the statistical analysis been performed appropriately and rigorously? 

Reviewer #1: I Don't Know

Reviewer #2: No

3. Have the authors made all data underlying the findings in their manuscript fully available?

Reviewer #1: No

Reviewer #2: No

4. Is the manuscript presented in an intelligible fashion and written in standard English?

Reviewer #1: No

Reviewer #2: No

5. Review Comments to the Author

Reviewer #1: Thank you for the invitation to review this paper.

This is a study on the effects of protozoa-positive fecal bacteriome transplantation (FBT) in germ-free mice on the immune system, comparing it to protozoa-negative bacteriome transplantation.

Fecal transplants were made from 14 individuals’ stool, of which half tested positive for protozoa. The study reports several findings, including a reduce in cytokines in the small intestine, such as IL-6, TNF, IFN-y, but also IL-10. Additionally, an increased mucus layer, an upregulation of genes involving the epithelial barrier, and a downregulation of Tregs cells were associated with protozoa-positive FBT.

The authors should be congratulated for pioneering research on protozoa in relation to the immune system via FBT in mice.

I do however have some concerns and suggestions to improve the article.

Major comments:

1. The authors argue that they applied a ‘fecal bacteriome transplantation’ instead of a fecal microbiota transplantation due to freezing of the stool samples, which would have eliminated the protozoa. No references are listed here. While freezing diminishes the viability of certain protozoa, such as Blastocystis hominis, it may not completely eliminate them after one freeze-thaw cycle.

See references: Hurych, J., Vodolanova, L., Vejmelka, J., Drevinek, P., Kohout, P., Cinek, O., & Nohynkova, E. (2022). Freezing of faeces dramatically decreases the viability of Blastocystis sp. and Dientamoeba fragilis. European journal of gastroenterology & hepatology, 34(2), 242-243.

And also Terveer, E. M., van Gool, T., Ooijevaar, R. E., Sanders, I. M., Boeije-Koppenol, E., Keller, J. J., ... & Kuijper, E. J. (2020). Human transmission of Blastocystis by fecal microbiota transplantation without development of gastrointestinal symptoms in recipients. Clinical infectious diseases, 71(10), 2630-2636. Where Blastocystis were transferred via frozen FMT.

Since the research question focuses on the influence of protozoa presence on the immune system, I suggest the authors consider extending the experiment with fresh FMT (without a freeze/thaw cycle) to provide more insights into direct and indirect influences of protozoa on the bacteriome, and consequently, the immune system.

If this is not possible or desirable, the limitations of current methods for (correlational or causal) interpretation should be discussed.

2. To better understand the reported results; it is important to consider following points for clarification:

* Provide information on the 7 participants who were protozoa positive; which were found and in what frequencies? The previous article (concerning 49 participants) were colonized with several protozoa Blastocystis hominis, Entamoeba coli, Entamoeba dispar, Endolimax nana, Iodamoeba butshlii, Giardia duodenalis, Chilomastix mesnili, Hymenolepis nana, and Ascaris lumbricoides.

* Include information on the bacteriome of these 14 participants; in that way the reader can interpret the results in the context of bacterial abundance and composition, which are argued to underlie the reported findings.

* Specify the number of FBTs conducted, 14 x 4 ?

3. A limitation of this study is that several protozoa (possibly 6?) have been studies in a small sample size. For example, both Blastocystis was studied, which is mostly considered as commensal, but also or maybe (this is not clear in the article) Gardia duodenalis and Entamoeba histolytica, which are known to have the capacity the cause substantial complaints. The variance in protozoa exposure in the stool could be a possible explanation for the seemingly contradictory results found in this research; such as the downregulation of pro-inflammatory markers, but also increased epithelial permeability. This limitation should be discussed.

4. The (clinical) relevance of this research is not explained in the introduction and should receive more attention.

Minor comments

1. The interpretation of reported findings and the implication of these findings are not clear. The authors should clarify how they interpret their findings.

2. Fecal microbiota transplantation and fecal bacteriome transplantation are inconsistently used throughout the article.

3. Consider to start the method section with a brief summary of the used cohort. It now starts with ‘every volunteer mother’, which is confusing since this cohort is not introduced. (line 97)

4. In lines 83-87 there appears a contradiction; it states that parasite positivity and parasite negativity are associated with increased abundance of Clostridia taxa.

5. The manuscript should be checked on spelling, grammar and clarity. (e.g. revise line 101-103, line 111-112).

6. Consider to delete line 151 since this sentence is stated twice.

7. Consider to rewrite line 344-346 to avoid redundancy with a section in the introduction.

8. In line 369 it is not clear that the findings of the authors contradict the statement in the previous line 368.

9. Add a reference for line 379-381 or clarify that this is speculation. The same applies for line 426 – 428.

10. The matching strategy should be explained (line. 137-138)

Reviewer #2: Review-Faecal bacteriome transplantation from protozoa-exposed donors downregulates immune response in a germ-free mouse model, its role in immune response and physiology of the intestine.

The study, a pioneer in its field, is the first to explore the role of a bacteriome shaped by the presence of intestinal protozoa and the use of FBT from protozoa-exposed individuals in a germ-free mouse model. The findings shed new light on the intricate biological processes, suggesting a potential regulatory and protective role of protists in maintaining intestinal homeostasis in a host.

The abstract needs to be extended.

The text is affected by linguistic errors.

The software and design are affected by a lot of irregularities.

The results are expected.

M and M need more extensive reports and descriptions.

The study design part needs to be more clearly described, and all the volunteers' details must be provided.

Species names must follow the citation style; once written, the next time, only the second names appear. Please correct this.

References: pl adapt to the journal's guidelines.

6. PLOS authors have the option to publish the peer review history of their article (what does this mean?). If published, this will include your full peer review and any attached files.

Reviewer #1: No

Reviewer #2: **Yes: **Professor Dr Panagiotis Karanis

---

## [Author Response · Author response to Decision Letter 0]

17 Aug 2024

Comments to the Author

1. Is the manuscript technically sound, and do the data support the conclusions?

Reviewer #1: Partly

Reviewer #2: Yes

Answer:

The reviewers' comments help us to make this revised version clearer and more robust. However, our work had some limitations. Nevertheless, we believe that the results obtained support our conclusions.

2. Has the statistical analysis been performed appropriately and rigorously? 

Reviewer #1: I Don't Know

Reviewer #2: No

Answer:

Regarding the analysis we used, it is necessary to clarify that it was carried out primarily based on the distribution of the data. We decided not to transform the data when a normal distribution was not met, since this could reduce the asymmetry and variability of the original data and could even increase them. When this happens, the estimation of the parameters of the models and the hypothesis tests can be affected. Therefore, we consider that the analyses used in this work are robust and are used to distinguish differences between populations.

3. Have the authors made all data underlying the findings in their manuscript fully available?

The PLOS Data policy requires authors to make all data underlying the findings described in their manuscript fully available without restriction, with rare exception (please refer to the Data Availability Statement in the manuscript PDF file). The data should be provided as part of the manuscript or its supporting information or deposited to a public repository. For example, in addition to summary statistics, the data points behind means, medians and variance measures should be available. If there are restrictions on publicly sharing data—e.g. participant privacy or use of data from a third party—those must be specified.

Reviewer #1: No

Reviewer #2: No

Answer:

The availability of the results and findings of this manuscript are included in supplementary data. However, it is important to mention that the volunteers signed informed consent for the use of their samples and associated data exclusively for this specific project. If further details are required, please contact to corresponding authors.

4. Is the manuscript presented in an intelligible fashion and written in standard English?

Reviewer #1: No

Reviewer #2: No

 Answer:

Before our first manuscript submission, we used an English editing service. Additionally, Brett Finlay Professor, a researcher and professor at British Columbia University in Canada, who is one of the authors, made the last revision of the manuscript. However, this new version was again carefully reviewed. We included the certificate of our English editing service. 

5. Review Comments to the Author

Reviewer #1: Thank you for the invitation to review this paper.

This is a study on the effects of protozoa-positive fecal bacteriome transplantation (FBT) in germ-free mice on the immune system, comparing it to protozoa-negative bacteriome transplantation.

Fecal transplants were made from 14 individuals’ stool, of which half tested positive for protozoa. The study reports several findings, including a reduce in cytokines in the small intestine, such as IL-6, TNF, IFN-y, but also IL-10. Additionally, an increased mucus layer, an upregulation of genes involving the epithelial barrier, and a downregulation of Tregs cells were associated with protozoa-positive FBT.

The authors should be congratulated for pioneering research on protozoa in relation to the immune system via FBT in mice.

I do however have some concerns and suggestions to improve the article.

Major comments:

1. The authors argue that they applied a ‘fecal bacteriome transplantation’ instead of a fecal microbiota transplantation due to freezing of the stool samples, which would have eliminated the protozoa. No references are listed here. While freezing diminishes the viability of certain protozoa, such as Blastocystis hominis, it may not completely eliminate them after one freeze-thaw cycle.

See references: Hurych, J., Vodolanova, L., Vejmelka, J., Drevinek, P., Kohout, P., Cinek, O., & Nohynkova, E. (2022). Freezing of faeces dramatically decreases the viability of Blastocystis sp. and Dientamoeba fragilis. European journal of gastroenterology & hepatology, 34(2), 242-243.

And also Terveer, E. M., van Gool, T., Ooijevaar, R. E., Sanders, I. M., Boeije-Koppenol, E., Keller, J. J., & Kuijper, E. J. (2020). Human transmission of Blastocystis by fecal microbiota transplantation without development of gastrointestinal symptoms in recipients. Clinical infectious diseases, 71(10), 2630-2636. Where Blastocystis were transferred via frozen FMT.

Since the research question focuses on the influence of protozoa presence on the immune system, I suggest the authors consider extending the experiment with fresh FMT (without a freeze/thaw cycle) to provide more insights into direct and indirect influences of protozoa on the bacteriome, and consequently, the immune system.

If this is not possible or desirable, the limitations of current methods for (correlational or causal) interpretation should be discussed.

 Answer:

Although we know that the suspension, we transplanted contains mainly bacterial populations, we cannot rule out the presence of parasitic products and some viral particles that could be participating in the events we explored. Therefore, we agree with the reviewer and change the term “fecal bacteriome transplantation” to “fecal microbiota transplantation” throughout the text of the article.

We revised the manuscripts suggested by our reviewer and added references to the treatment of fecal samples in fecal microbiota transplants in the material and methods section (Lines 210-217). 

Since it was not initially proposed in the experimental design to use fresh fecal samples, we couldn't add FMT information with fresh samples to analyze the direct influence of parasites, for this reason, we discuss this as a limitation of our work (Lines 406-413).

2. To better understand the reported results; it is important to consider the following points for clarification:

* Provide information on the 7 participants who were protozoa positive; which were found and in what frequencies? The previous article (concerning 49 participants) were colonized with several protozoa Blastocystis hominis, Entamoeba coli, Entamoeba dispar, Endolimax nana, Iodamoeba butshlii, Giardia duodenalis, Chilomastix mesnili, Hymenolepis nana, and Ascaris lumbricoides.

* Include information on the bacteriome of these 14 participants; in that way the reader can interpret the results in the context of bacterial abundance and composition, which are argued to underlie the reported findings.

Answer:

In the material and methods section, we describe the general characteristics of all participants (Lines 163-169 and Lines 174-180). 

In addition, we include which parasites were found in the multi-parasitized samples and their frequencies (Lines 161-171). 

We also include information on the composition and abundance of the bacteriome of the 14 participants (Lines 181-185). We also specify the number of FMT repetitions performed (Lines 221-226). 

Additionally, we add more detailed information on these characteristics in supplementary material.

3. A limitation of this study is that several protozoa (possibly 6?) have been studied in a small sample size. For example, both Blastocystis was studied, which is mostly considered as commensal, but also or maybe (this is not clear in the article) Gardia duodenalis and Entamoeba histolytica, which are known to have the capacity the cause substantial complaints. The variance in protozoa exposure in the stool could be a possible explanation for the seemingly contradictory results found in this research, such as the downregulation of pro-inflammatory markers, but also increased epithelial permeability. This limitation should be discussed. 

Answer:

In this study, we worked only with individuals who did not present signs or symptoms related to intestinal disease (Lines 179-181). We can advance here that asymptomatic multi-parasitic infection is based on the clinical diagnosis, however, we cannot rule out subclinical tissue damage (increased epithelial permeability) due to the downregulation of pro-inflammatory immune response. Further, we worked with multi-parasitized samples and did not discriminate between potentially pathogenic and non-pathogenic organisms. Therefore, this variation in parasite composition could generate variation in the results. We signal this limitation in the discussion (Lines 402-405).

4. The (clinical) relevance of this research is not explained in the introduction and should receive more attention.

Answer:

We agree with the reviewer, so we elaborate on the clinical relevance of our study in the introduction (lines 133-142).

Minor comments

1. The interpretation of reported findings and the implication of these findings are not clear. The authors should clarify how they interpret their findings.

Answer:

We made changes in this regard in the discussion and conclusions sections to be sufficiently clear in the revised version.

2. Fecal microbiota transplantation and fecal bacteriome transplantation are inconsistently used throughout the article.

 Answer:

We correct this inconsistency in the revised manuscript.

3. Consider to starting method section with a summary of the used cohort. It now starts with ‘every volunteer mother’, which is confusing since this cohort is not introduced. (line 97)

 Answer:

 We followed the reviewer's suggestion and added a summary of the cohort studied (Lines 152-185).

4. In lines 83-87 there appears a contradiction; it states that parasite positivity and parasite negativity are associated with increased abundance of Clostridia taxa.

We corrected this paragraph (L123-L132). Although the Clostridia taxa were one of the most abundant groups, both in parasitized and non-parasitized individuals, the abundance was significantly higher in the group of parasitized individuals.

5. The manuscript should be checked on spelling, grammar and clarity. (e.g. revise line 101-103, line 111-112).

Answer:

The manuscript was reviewed by the editing service (we include the certificate). 

6. Consider to delete line 151 since this sentence is stated twice.

 Answer:

The repeated line was deleted. 

7. Consider to rewrite line 344-346 to avoid redundancy with a section in the introduction.

 Answer:

 This suggestion was now included (lines 398-401). 

8. In line 369 it is not clear that the findings of the authors contradict the statement in the previous line 368.

Answer:

We rewrote this paragraph to make it clear that other studies have shown that non-pathogenic organisms decrease proinflammatory cytokine concentrations, as found in our results (Lines 425-443).

9. Add a reference for line 379-381 or clarify that this is speculation. The same applies for line 426 – 428.

 Answer: 

 We change “This situation suggests” to “We speculate” (Lines 436-439).

Lines 426-428 were deleted.

10. The matching strategy should be explained (line. 137-138)

 Answer:

This suggestion is added in the material and methods section (Lines 206-209). 

Reviewer #2: Review-Faecal bacteriome transplantation from protozoa-exposed donors downregulates immune response in a germ-free mouse model, its role in immune response and physiology of the intestine.

The study, a pioneer in its field, is the first to explore the role of a bacteriome shaped by the presence of intestinal protozoa and the use of FBT from protozoa-exposed individuals in a germ-free mouse model. The findings shed new light on the intricate biological processes, suggesting a potential regulatory and protective role of protists in maintaining intestinal homeostasis in a host.

The abstract needs to be extended.

The text is affected by linguistic errors.

Answer:

We restructured the summary to make it more understandable. 

This version was carefully revised. We included the certificate of the English edition.

The software and design are affected by a lot of irregularities.

The results are expected.

Answer: 

Our apologies to our reviewer, we couldn't understand your question.

M and M need more extensive reports and descriptions.

The study design part needs to be more clearly described, and all the volunteers' details must be provided.

 Answer:

We have carried out a detailed revision of this section. The revised version includes all information about our volunteers in the material and methods section and adds files as supplementary information.

Species names must follow the citation style; once written, the next time, only the second names appear. Please correct this.

References: pl adapt to the journal's guidelines.

 Answer. 

The revised version was carefully reviewed and species names, citation style, and references were corrected.

---

## [Decision Letter · Decision Letter 1]

8 Sep 2024

PONE-D-24-07909R1Fecal microbiota transplantation from protozoa-exposed donors downregulates immune response in a germ-free mouse model, its role in immune response and physiology of the intestine.PLOS ONE

Dear Dr. Ximenez,

Thank you for submitting your manuscript to PLOS ONE. After careful consideration, we feel that it has merit but does not fully meet PLOS ONE’s publication criteria as it currently stands. Therefore, we invite you to submit a revised version of the manuscript that addresses the points raised during the review process.

It appears that most of the scientific concerns have been adequately addressed. However, there remain a substantial number of clerical, typographical, and logistical errors and disorganization in the manuscript. Please carefully edit the manuscript, paying particular attention to those issues noted by the reviewer.

We look forward to receiving your revised manuscript.

Kind regards,

Brenda A Wilson, Ph.D.

Academic Editor

PLOS ONE

Journal Requirements:

Reviewers' comments:

Reviewer's Responses to Questions

**Comments to the Author**

1. If the authors have adequately addressed your comments raised in a previous round of review and you feel that this manuscript is now acceptable for publication, you may indicate that here to bypass the “Comments to the Author” section, enter your conflict of interest statement in the “Confidential to Editor” section, and submit your "Accept" recommendation.

Reviewer #2: All comments have been addressed

2. Is the manuscript technically sound, and do the data support the conclusions?

Reviewer #2: Yes

3. Has the statistical analysis been performed appropriately and rigorously? 

Reviewer #2: No

4. Have the authors made all data underlying the findings in their manuscript fully available?

Reviewer #2: Yes

5. Is the manuscript presented in an intelligible fashion and written in standard English?

Reviewer #2: Yes

6. Review Comments to the Author

Reviewer #2: Fecal microbiota transplantation from protozoa-exposed donors downregulates

immune response in a germ-free mouse model, its role in immune response and

physiology of the intestine.

Review

l. 429 pl don’t write spp in cursive

2. 485, 486 please don’t write phylum and families in cursive

3. References

Your refs are still affected with many irregularities and citation style mistakes

Examples below- first 4 references !!!!! (but also subsequently)

1. Partida-Rodriguez O, Serrano-Vazquez A, Nieves-Ramirez M, Moran P, Rojas L, Portillo T, et al.3

Human Intestinal Microbiota: Interaction Between Parasites and the Host Immune Response. Arch4

Med Res. 2017;48: 690-700. doi: 10.1016/j.arcmed.2017.11.01556

2. Parfrey LW, Walters WA, Knight R. Microbial Eukaryotes in the Human Microbiome: Ecology,7

Evolution, and Future Directions. Front Microbiol. 2011;2: 1-6. doi: 10.3389/fmicb.2011.00153.8

3. Scanlan PD, Marchesi JR. Micro-eukaryotic diversity of the human distal gut microbiota:9

qualitative assessment using culture-dependent and -independent analysis of feces. SME J. 2008;2:10

1183-1193. doi: 10.1038/ismej.2008.76.11

4. Aboulhoda BE, Abdelfatah M, El-Wakil ES, Alghamdi M, Albadawi EA, et al. Microbiota-12

Parasite Interaction: Implication of Secretory Immunoglobulin A and P2X7 Receptor Signaling.13

Discov Med. 2024 Feb;36(181):217-233. doi: 10.24976/Discov.Med.202436181.21. PMID:14

38409828.15

5. Verma AK, Verma R, Ahuja V, Paul J. Real-time analysis of gut flora in Entamoeba histolytica16

infected patients of Northern India. BMC Microbiol. 2012;12: 1-11 doi: 10.1186/1471-2180-12-183.

7. PLOS authors have the option to publish the peer review history of their article (what does this mean?). If published, this will include your full peer review and any attached files.

Reviewer #2: No

---

## [Author Response · Author response to Decision Letter 1]

24 Sep 2024

Response to reviewers (PONE-D-24-07909R1)

Reviewer #2: Fecal microbiota transplantation from protozoa-exposed donors downregulates immune response in a germ-free mouse model, its role in immune response and physiology of the intestine.

Questions 

1. In line 429 please don't write spp in cursive

Answer: We removed the italics (now in line 436).

2. In lines 485, 486 please don't write phylum and families in cursive

Answer: We removed italics in phylum and families (now in lines 492 and 493).

3. References

Your references are still affected with many irregularities and citation style mistakes

Answer: We have reviewed and corrected every reference, considering the parameters suggested by PLOS ONE.

---

## [Editor Report · Decision Letter 2]

14 Oct 2024

Fecal microbiota transplantation from protozoa-exposed donors downregulates immune response in a germ-free mouse model, its role in immune response and physiology of the intestine.

PONE-D-24-07909R2

Dear Dr. Ximenez,

We’re pleased to inform you that your manuscript has been judged scientifically suitable for publication and will be formally accepted for publication once it meets all outstanding technical requirements.

Kind regards,

Brenda A Wilson, Ph.D.

Academic Editor

PLOS ONE
---

## [Editor Report · Acceptance letter]

16 Oct 2024

PONE-D-24-07909R2 

PLOS ONE

Dear Dr. Ximenez, 

I'm pleased to inform you that your manuscript has been deemed suitable for publication in PLOS ONE. Congratulations! Your manuscript is now being handed over to our production team.

Kind regards, 

on behalf of

Dr. Brenda A Wilson 

Academic Editor

PLOS ONE